# CRISPR/Cas9-Mediated *HY5* Gene Editing Reduces Growth Inhibition in Chinese Cabbage (*Brassica rapa*) under ER Stress

**DOI:** 10.3390/ijms241713105

**Published:** 2023-08-23

**Authors:** Ye Rin Lee, Ki Seong Ko, Hye Eun Lee, Eun Su Lee, Koeun Han, Jae Yong Yoo, Bich Ngoc Vu, Ha Na Choi, Yoo Na Lee, Jong Chan Hong, Kyun Oh Lee, Do Sun Kim

**Affiliations:** 1Vegetable Research Division, National Institute of Horticultural and Herbal Science, Rural Development Administration, Wanju-gun 55365, Republic of Korea; lyr1219@korea.kr (Y.R.L.); helee72@korea.kr (H.E.L.); lus4434@korea.kr (E.S.L.); hke1221@korea.kr (K.H.); 2Plant Molecular Biology and Biotechnology Research Center (PMBBRC), Gyeongsang National University, 501 Jinju-daero, Jinju 52828, Republic of Korea; bug0@nate.com (K.S.K.); jayong77@naver.com (J.Y.Y.); jchong@gnu.ac.kr (J.C.H.); 3Division of Life Science, Division of Applied Life Sciences (BK4 Program), Gyeongsang National University, 501 Jinju-daero, Jinju 52828, Republic of Korea; vungock57cnshe@gmail.com (B.N.V.); chn7514@naver.com (H.N.C.); dbsksp99@naver.com (Y.N.L.)

**Keywords:** HY5, ER stress, NBTs, UPR, gene editing, Chinese cabbage

## Abstract

Various stresses can affect the quality and yield of crops, including vegetables. In this study, CRISPR/Cas9 technology was employed to examine the role of the *ELONGATED HYPOCOTYL 5* (*HY5*) gene in influencing the growth of Chinese cabbage (*Brassica rapa*). Single guide RNAs (sgRNAs) were designed to target the *HY5* gene, and deep-sequencing analysis confirmed the induction of mutations in the bZIP domain of the gene. To investigate the response of Chinese cabbage to endoplasmic reticulum (ER) stress, plants were treated with tunicamycin (TM). Both wild-type and *hy5* mutant plants showed increased growth inhibition with increasing TM concentration. However, the *hy5* mutant plants displayed less severe growth inhibition compared to the wild type. Using nitroblue tetrazolium (NBT) and 3,3′-diaminobenzidine (DAB) staining methods, we determined the amount of reactive oxygen species (ROS) produced under ER stress conditions, and found that the *hy5* mutant plants generated lower levels of ROS compared to the wild type. Under ER stress conditions, the *hy5* mutant plants exhibited lower expression levels of UPR- and cell death-related genes than the wild type. These results indicate that CRISPR/Cas9-mediated editing of the HY5 gene can mitigate growth inhibition in Chinese cabbage under stresses, improving the quality and yield of crops.

## 1. Introduction

Crops such as rapeseed, along with vegetables like cabbage and Chinese cabbage, all of which belong to the Brassica genus, are widely cultivated across the world. Chinese cabbage, in particular, holds significant commercial importance as a vegetable extensively consumed in Asian countries [1,2,3]. Chinese cabbage thrives well in optimal temperatures ranging from 15 to 20 °C. Aberrant high-temperature conditions can lead to growth inhibition, as well as yellowing and wilting of leaves. In extreme cases, these factors may culminate in the destruction of leaf heads, which increases vulnerability to infectious diseases and subsequently leads to significant reductions in both quality and yield [3,4,5]. Heat stress can exert a pronounced influence during the transition of Chinese cabbage from the vegetative to the reproductive growth stage [6,7]. Besides high temperature, Chinese cabbage is also susceptible to various environmental stressors such as drought, salinity, herbicide application, and pathogen infection, all of which detrimentally impact its quality and yield. The increasing use of fossil fuels, which leads to global warming and climate change, is progressively exacerbating conditions for the cultivation of crops and vegetables. Consequently, there is an urgent need to develop new and innovative mechanisms to protect plants from these adverse cultivation conditions [3,5,8,9,10].

Environmental stresses such as high temperatures, drought, salinity, herbicide application, and pathogen infection can impair the protein folding capacity of the endoplasmic reticulum (ER) in plants, resulting in ER stress [11,12,13]. ER stress substantially affects both the quality and yield of various crops and vegetables, including Chinese cabbage [3,9,10]. In response to ER stress, plant cells have evolved an adaptive cellular mechanism called the unfolded protein response (UPR). Under ER stress conditions, cells strive to maintain ER homeostasis through the transcriptional induction of ER chaperones, the transient attenuation of global protein synthesis, and ER-associated protein degradation (ERAD) [10,11,12,14]. However, prolonged ER stress can trigger pro-apoptotic signals to protect the organism from rogue cells that are filled with irreversibly denatured proteins and toxic molecules [10,11,12,15]. Tunicamycin (TM) is a natural antibiotic that induces ER stress in cells by hindering the early phase of N-linked glycan synthesis in proteins. This interference results in the buildup of misfolded proteins, triggering the UPR in plants [16].

The UPR, a conserved mechanism in eukaryotic organisms including plants, is pivotal for cellular adaptation to stress [10,11,15]. It acts as a key regulatory pathway in plant responses to combined biotic and abiotic stresses, equipping plants to handle environmental challenges. Increased expression of ER chaperones, such as BiP (GRP78), calreticulin, and calnexin, along with transcriptional upregulation of UPR genes, is commonly linked to elevated ER stress response [11,17,18,19]. In plants, two ER stress sensors, IRE1 and bZIP28, play an essential role in detecting the accumulation of unfolded proteins in the ER and transmitting signals to intracellular components to induce the UPR. Upon ER stress, IRE1 splices the unprocessed bZIP60u mRNA to generate the spliced form, bZIP60s. This spliced form subsequently translocates to the nucleus to bind to *cis*-elements located in the promoters of UPR genes [20,21]. Concurrently, bZIP28, a membrane-bound transcription factor, responds to ER stress by releasing its cytosolic portion, which contains the bZIP domain, from the membrane via proteolytic cleavage. This portion then translocates to the nucleus, where it induces transcription of UPR genes. These two plant UPR pathways work in conjunction to constitute the stress transcriptome, modulating gene expression and sequentially regulating various biochemical and physiological processes in response to ER stress [20,21].

The ELONGATED HYPOCOTYL 5 (HY5), a bZIP transcription factor, plays a central role in plant photomorphogenesis, participating in various developmental processes including hypocotyl elongation, hormone regulation, and anthocyanin synthesis [22,23,24,25,26,27,28,29,30]. HY5 also regulates the expression of nearly one-third of the genes in *Arabidopsis thaliana* [22,28,30,31,32]. Conversely, COP1, which interacts with HY5, serves as a key negative regulator in light signaling, repressing photomorphogenesis in the dark [31,33]. As an E3 ubiquitin ligase, COP1 binds to and degrades HY5 in the dark, preventing its interaction with protein partners or binding to downstream promoters, thereby resulting in elongated hypocotyls [34,35]. HY5 orchestrates hormones such as gibberellins, brassinosteroids, abscisic acid, ethylene, and auxin to promote photomorphogenesis [13,23,24,25,27,30,36]. Previous research has suggested that light induces the UPR in conjunction with abiotic stresses and that the UPR pathway can be integrated with the light signaling pathway [37]. Previous research has also demonstrated that mutations in HY5 can alleviate sensitivity to ER stress in Arabidopsis [38], but so far, no reports have addressed the impact of HY5 on stress responses in Chinese cabbage.

Contemporary research is largely focused on the development of crops and vegetables with traits that enable them to overcome adverse environmental stresses. While these techniques traditionally involve T-DNA insertion into the plant genome, advancements in molecular biology, particularly the advent of clustered regularly interspaced short palindromic repeats (CRISPR) and the associated system (CRISPR/Cas), now facilitate the manipulation of plant characteristics without necessitating T-DNA insertion [39]. Utilizing the CRISPR-Cas genome editing system, significant progress has been made in plant genetic optimization. Analogous to textual refinement, this technique enables profound phenotypic modifications in crops. For instance, targeted alterations in the GT-1 element of rice *RAV2* gene enhanced salinity tolerance, emphasizing the pivotal role of the GT-1 [40,41]. In maize, modifications in the *ARGOS8* gene, linked to ethylene response, resulted in increased grain yield under suboptimal hydration conditions [42,43,44]. Furthermore, a specific edit in rice *SWEET11* gene augmented its disease resistance without compromising fecundity [45]. Another genetic intervention bolstered rice resistance to bacterial blight [46]. These revelations illuminate the potential of genome editing in cultivating resilient and high-yielding crops. In this context, our study employs CRISPR/Cas9 technology to introduce mutations in the bZIP domain of the *HY5* gene in Chinese cabbage. We then compared the growth of these mutant plants under ER stress conditions with that of the wild type. Our results suggest that CRISPR/Cas9-mediated editing of the *HY5* gene can potentially alleviate growth inhibition in Chinese cabbage caused by various environmental stresses. This finding is a significant step towards improving the quality and yield of this and potentially other crops and vegetables.

## 2. Results and Discussion

### 2.1. Introduction of Mutations in the HY5 Gene Using CRISPR/Cas9 Technology for HY5-Deficient Chinese Cabbage Mutants

New breeding techniques (NBTs), such as genome editing, enable breeders to introduce specific desirable traits, like disease resistance or stress tolerance, into plants more precisely and rapidly compared to traditional breeding methods. Known for its role in photomorphogenesis, HY5 has also been found to participate in responses to various environmental stresses such as high salinity, drought, and cold [23,47,48,49]. Moreover, it is implicated in hormone signaling pathways, including those of auxin, gibberellin, and abscisic acid, as well as in the response to ER stress [13,23,24,25,27,30,36]. In this study, we induced mutations in the *HY5* gene using CRISPR/Cas9 technology and generated Chinese cabbage mutants lacking HY5 function. By examining their response to ER stress, we explored the potential value of the *HY5* gene as a novel target for NBTs, aimed at developing crops and vegetables capable of coping with environmental stresses. To design the guide RNAs used to create Chinese cabbage mutants lacking HY5 function using CRISPR/Cas9 technology, we performed a BRAD BLAST search using the sequence of the *HY5* gene (At5G11260) from Arabidopsis. This resulted in the identification of the *Brassica rapa HY5* gene (Bra008976), which displayed a 90% sequence similarity to the *HY5* gene of Arabidopsis. It is well-known that the HY5 protein consists of a bZIP domain and a COP1 interacting domain to play its role in light signal transduction and other cellular processes (Figure 1A). Two sgRNAs were designed to target sequences within the bZIP domain, which is necessary for DNA binding (Figure 1A,B). These were then ligated into pAGM4723, resulting in the construction of the CRISPR/Cas9 vector (Figure 1C). The CRISPR/Cas9 vector was introduced into Agrobacterium GV3101 to facilitate plant transformation. Approximately 3000 hypocotyl pieces from the ‘Seoul’ cultivar of Chinese cabbage were used in the creation of Chinese cabbage mutants lacking HY5 function (Figure 1D). After a two-week co-cultivation period with the Agrobacterium, around 40–50% of the hypocotyls began callus formation at the cut ends. Adventitious shoots sprouted from these calli over the following 3–4 weeks (Figure 1D). The shoots were subsequently separated from the callus and relocated to a regeneration medium (Figure 1D). After regeneration, the shoots were transferred to a rooting medium to facilitate root formation. Finally, the fully regenerated plants underwent a two-day acclimatization process in the rooting medium prior to their transplantation into pots filled with soil (Figure 1D).

### 2.2. CRISPR/Cas9-Mediated Gene-Editing Results in Efficient Loss-of-Function HY5 Mutations in Transgenic Chinese Cabbage

As a result of transformation using the CRISPR/Cas9 vector, a total of 22 regenerated Chinese cabbage lines were obtained. First, the gDNA was isolated from these regenerated Chinese cabbage lines, and the insertion of T-DNA including sgRNAs was confirmed via PCR using a pair of sgRNA-specific primer sets. Among the regenerated 22 Chinese cabbage lines, 19 lines were identified to contain the anticipated 300 bp fragment. This result indicates successful transformation with the T-DNA, including sgRNAs (Figure 2A). Next, further examination of the mutations induced by the two designed sgRNAs and CRISPR/Cas9 and the evaluation of gene-editing efficiency were conducted in the 19 lines with confirmed T-DNA insertions. For this purpose, the gDNA was isolated from these lines, PCR was performed using a pair of *HY5* gene-specific primer sets, and the resulting PCR products were subjected to targeted deep sequencing analysis. Analysis results confirmed that all transformed lines had successfully edited sequences, generating mutations via sgRNAs and CRISPR/Cas9. 

Every independent transformed Chinese cabbage line produced two types of gene editing patterns, each induced by sgRNA1 and sgRNA2 (Figure 2B). The editing pattern observed for sgRNA1 was either the deletion of three nucleotides or the insertion of one nucleotide, while the editing pattern observed for sgRNA2 was the deletion of two or three nucleotides (Figure 2B). The majority of editing events by sgRNA occurred approximately 3 bp upstream of the protospacer adjacent motif (PAM) sequence, as predicted for Cas9-mediated DNA cleavage [50]. To assess the efficiency of gene editing, *E2* and *E10* lines were chosen to investigate the editing frequency of each mutation type. The *E2* line showed total editing efficiencies of 95.6% and 92.1% for sgRNA1 and sgRNA2, respectively (Table 1). Editing patterns with three nucleotide deletions and one nucleotide insertion by sgRNA1 were observed in 62.3% and 33.3% of cases, respectively (Table 1). Editing patterns with two nucleotide deletions and three nucleotide deletions by sgRNA2 were observed in 57.5% and 34.6% of cases, respectively (Table 1). The *E10* line showed total editing efficiencies of 97.3% and 97.1% for sgRNA1 and sgRNA2, respectively. Editing patterns with three nucleotide deletions and one nucleotide insertion by sgRNA1 were observed in 62.2% and 35.1% of cases, respectively (Table 1). Editing patterns with two nucleotide deletions and three nucleotide deletions by sgRNA2 were observed in 56.3% and 40.8% of cases, respectively (Table 1). Based on the results of deep sequencing analysis, lines with three nucleotide deletions that caused the loss of one amino acid but maintained the reading frame were excluded from further analyses. On the other hand, lines with patterns of one nucleotide insertion or two nucleotide deletions causing frame shift indel mutations were selected for further analyses. These lines are predicted to induce premature termination of mRNA translation and produce HY5 proteins with loss of function (Figure 2C). These results demonstrate that CRISPR/Cas9-induced *HY5* gene editing can efficiently induce indel mutations and can be used to generate Chinese cabbage with loss-of-function HY5 protein.

### 2.3. Mutations in the HY5 Gene Confer Resistance to Tunicamycin-Induced ER Stress in Chinese Cabbage

It is well-documented that ER stress can impact the quality and yield of various crops and vegetables, including Chinese cabbage [3,9,29]. To investigate the impact of ER stress on the growth of Chinese cabbage, sterilized seeds were grown on MS agar medium containing varying concentrations of TM for 5 d, and their growth and fresh weight were compared (Figure 3A,B). Chinese cabbages grown on MS agar medium containing 50 and 100 ng/mL of TM showed mild growth inhibition and a corresponding decrease in fresh weight compared to those grown on TM-free medium (Figure 3A,B). However, Chinese cabbages grown on a medium containing 200 ng/mL or more of TM showed severe growth inhibition and a significant decrease in fresh weight (Figure 3A,B). This result indicates that ER stress induced by TM can negatively affect the growth of Chinese cabbage, which appears to have higher tolerance to ER stress induced by TM at concentrations typically inhibiting growth in Arabidopsis (over 20 ng/mL) [18,19]. To investigate whether the *HY5* gene mutation in Chinese cabbage confers resistance to TM-induced ER stress, wild-type and *hy5* mutant Chinese cabbage lines, edited with CRISPR/Cas9, were grown on MS agar medium without TM and with 150 and 200 ng/mL TM for 7 d, and their phenotypes and fresh weight decreases were compared (Figure 3C,D). In the absence of TM, both *E2* and *E10* mutant cabbage lines exhibited similar growth and fresh weight to the wild type (Figure 3C,D). However, when grown in medium with 150 and 200 ng/mL TM, these mutant lines displayed a noticeable decrease in growth inhibition and a corresponding attenuation in fresh weight loss (Figure 3C,D). This result suggests that the *HY5* mutations mediated by CRISPR/Cas9 gene editing can confer resistance to ER stress in a major vegetable crop like Chinese cabbage.

### 2.4. Mutations in the HY5 Gene Reduce ROS Generation in Chinese Cabbage

ROS are important signaling molecules involved in many cellular processes, including stress responses. Normally, a balance is maintained between ROS generation and their removal by antioxidant defense systems, but this balance can be disrupted under stressful conditions, leading to ROS accumulation, which, if uncontrolled, can result in oxidative stress and cell death [51,52]. The impact of HY5 mutations on ROS levels in Chinese cabbage under TM-induced ER stress was examined. Wild-type and *E2* and *E10* mutant lines were grown on MS agar medium without TM and with 150 ng/mL TM for 7 d, and the generation and levels of H_2_O_2_ and O_2_^−^ were compared using DAB and NBT staining methods. Without TM, the mutant lines displayed DAB and NBT staining intensities similar to the wild type (Figure 4A,B). However, in the presence of 150 ng/mL TM, these mutant lines showed reduced staining intensities (Figure 4A,B). This result indicates that the *HY5* mutations mediated by CRISPR/Cas9 gene editing can reduce ROS production and thus stress tolerance in Chinese cabbage under ER stress.

### 2.5. Alleviation of Growth Inhibition in hy5 Mutants under ER Stress Induced by TM Is Associated with Decreased Expression of UPR- and Cell-Death-Related Genes

Previous research has reported increased expression of UPR- and cell-death-related genes in Arabidopsis under prolonged ER stress conditions [17,18,19,53,54]. These genes play a pivotal role in regulating various physiological processes essential for plant growth and development. Specifically, UPR-related genes, including *BiP2*, *BiP3*, *PDIL1-1*, *PDIL2-2*, *CNX1*, and *CRT3*, are primarily responsible for maintaining cellular homeostasis. They work to ensure that proteins are correctly folded and functionally active, thereby preventing the aggregation of misfolded proteins that can be detrimental to the cell. Similarly, cell-death-related genes like *BI-1* and *SIB1* play a crucial role in programmed cell death, an essential process in plant development and defense against pathogens. To understand the link between the alleviation of growth inhibition in *hy5* mutants under ER stress induced by TM and the expression level of UPR and cell death genes, we analyzed their expression using qRT-PCR. Wild-type and *E2* and *E10* mutant lines were grown on MS agar medium both without TM and with 150 ng/mL TM for 7 d. Under the no TM conditions, the mutant lines showed comparable UPR- and cell-death-related gene expression levels to the wild type (Figure 5). However, when exposed to 150 ng/mL TM, these mutant lines displayed diminished expression of the UPR-related genes (*BiP2*, *BiP3*, *PDIL2-2*, and *CRT3*) and cell-death-related genes (*BI-1* and *SIB1*) compared to the wild type (Figure 5). These findings suggest that the *HY5* mutations introduced via CRISPR/Cas9 gene editing can confer stress resistance to Chinese cabbage. This resistance seems to be linked to a decrease in the expression of UPR- and cell-death-related genes in Chinese cabbage under ER stress.

## 3. Materials and Methods

### 3.1. Plant Materials and Growth Conditions

The Chinese cabbage (*Brassica rapa*) cultivar ‘Seoul’ was utilized in this study for the development of edited plants using Agrobacterium-mediated transformation. The seed surfaces were submerged in 70% ethanol for 1 to 2 min, followed by agitation in a 2% sodium hypochlorite (NaOCl) solution for 10 min at 100 rpm. Subsequently, the seeds were rinsed thoroughly four to five times with sterilized water to remove any remaining NaOCl. The seeds were then germinated on ½ Murashige and Skoog (MS) medium (Duchefa Biochemie BV, Haarlem, Netherlands), containing 2% sucrose and 0.8% agar, and grown in vitro until the hypocotyls reached a length of 6–7 cm over five days at 24 °C under long day conditions (16/8 h light/dark photoperiod, 100–200 μmol/m^2^/s photon flux density, and 60–70% relative humidity). The hypocotyls were subsequently trimmed into 7–8 mm segments, carefully avoiding the shoot apex. These hypocotyl explants were used for Agrobacterium-mediated gene editing.

### 3.2. Sequence Information and Analysis

The gene and protein sequence information utilized in this study was sourced from the National Center for Biotechnology Information (NCBI, https://www.ncbi.nlm.nih.gov, accessed on 25 March 2021) and the Brassicaceae Database (BRAD, http://brassicadb.cn, accessed on 26 March 2021). The gene sequences of interest for Arabidopsis were retrieved from the NCBI GenBank. Subsequently, sequence similarities were detected, and gene sequence information for the corresponding genes in *Brassica rapa* was obtained using BRAD BLAST.

### 3.3. Selection of Target Sequences and Vector Construction

A region within the bZIP domain of the HY5 gene that includes a PAM sequence was selected as the target for our sgRNAs. For the filtering of false-positive results, Cas-OFFinder (http://www.rgenome.net/cas-offinder/, accessed on 1 April 2021) was utilized to predict the likelihood of potential off-target effects. The RNA Folding Form (http://www.unafold.org/mfold/applications/rna-folding-form.php, accessed on 1 April 2021) was employed to ensure the prediction of the potential for stem-loop formation, ensuring its absence. Based on these predictions, appropriate sgRNAs were synthesized for further experimentation. Vectors were assembled using the ‘Golden Gate’ modular cloning method [55]. To generate the sgRNA expression cassettes, DNA fragments containing sgRNA and a backbone were amplified using primers flanked with *Bsa*I restriction sites associated with ‘Golden Gate’-compatible overhangs. The amplicons were then assembled with the U6 promoter (pICSL01009) in Level 1 vectors pICH47751 and pICH47761, following the ‘Golden Gate’ protocol with the *Bsa*I-HF enzyme. Combinations of four Level 1 vectors, containing a hygromycin resistance selectable marker (pICSL11059::35SP:hptII), a Cas9 expression cassette (pICH47742::pRPS5a:pcoCas9), an sgRNA expression cassette (pICH47751::U6p:gRNA1 and pICH47761::U6p:gRNA2), and a linker cassette (pICH41780) were assembled into the Level 2 vector pAGM4723 via the ‘Golden Gate’ protocol using the *Bpi*I enzyme. All the plasmids were prepared using a GeneAll miniprep kit with *Escherichia coli* XL1-blue competent cells, selected with appropriate antibiotics and color selection. Finally, the resulting plasmid was introduced into the Agrobacterium strain GV3101 via electroporation.

### 3.4. Agrobacterium-Mediated Transformation and Selection Procedures

A single colony of the Agrobacterium strain GV3101, carrying the target construct, was cultured overnight in 10 mL of YEP liquid medium supplemented with hygromycin (50 mg/mL), at 28 °C and 250 rpm, until it reached an OD600 value of 0.8. The Agrobacterium suspension was centrifuged at 13,000 rpm for 10 min at 4 °C, yielding a pellet that was thoroughly washed and resuspended in MS medium containing 3.6% glucose for 1 h. On a clean bench, the pre-cultured explants were immersed in the Agrobacterium suspension for 10 min, and subsequently transferred to sterile filter paper to drain excess suspension. After infection, the explants were cultivated on MS medium supplemented with 3% sucrose, BA (4 mg/L), NAA (1 mg/L), and 0.8% plant agar (pH 5.8). They were then placed under dark conditions at 22 °C for 3 d. After co-cultivation, the explants were washed with MS liquid medium supplemented with cefotaxime (200 mg/L) and transferred to MS selection medium containing 3% sucrose, IBA (4 mg/L), NAA (3 mg/L), AgNO_3_ (4 mg/L), acetosyringone (5 mg/L), cefotaxime (200 mg/L), hygromycin (10 mg/L), and 0.8% plant agar (pH 5.6). After cultivation periods of 3 and 8 weeks, the formation of calli and shoots on the explants was recorded, respectively. Finally, the regenerated shoots were transferred to a rooting medium containing half-strength MS medium, 3% sucrose, cefotaxime (200 mg/L), and 0.7% plant agar (pH 5.8).

### 3.5. Mutation Detection and Deep Sequence Analysis

Prior to acclimation, genomic DNA (gDNA) was extracted from leaf samples of potential transgenic plants utilizing the cetyltrimethylammonium bromide (CTAB) method [56]. The gDNA was then quantified using a NanoDrop spectrophotometer (Nanodrop Technology). The status of these potential transgenic plants was assessed using PCR with a T-DNA-specific primer pair (tHSP-3UTer-F: 5′-GCTTGTTGTGTTATGAATTTGTGGC-3′ and RB-R: 5′-CAAACCGGCCAGGATTTCATG-3′). PCR conditions were set to 94 °C for 5 min, followed by 30 cycles of denaturation at 94 °C for 30 s, annealing at 58 °C for 30 s, and extension at 72 °C for 30 s, with a final extension at 72 °C for 5 min. The resulting PCR products were confirmed through 1.3% agarose gel electrophoresis and visualized with Evergreen dye under UV light. Transgenic plants confirmed to have T-DNA insertion via gel electrophoresis were then subjected to deep sequence analysis.

Samples that indicated target site insertion were also subjected to deep sequence analysis. For this deep sequencing analysis, three rounds of PCR were performed. The first round of PCR was used for DNA fragmentation, with primers designed to yield a product size of 600–800 bp (1st F: 5′-AAGCAGCGAGAGATCCTCAAG-3′ and 1st R: 5′-ATGAGCTATTCCAAGAACCACTGA-3′). The second round of PCR was used to attach the adapter, with primers designed to yield a product size of 150–250 bp (Adapt F: 5′-CGGTTAGAAGGTAGGTAATTCGG-3′, and Adapt R: 5′-CATCTGGTTCTCGTTCTGCAA-3′). The first and second rounds PCR were performed in a final volume of 20 μL, containing a 50 ng template DNA concentration, 10 pmol each of reverse and forward primers, 10 mM dNTPs, 10 × Hipi buffer, and 0.2 units of Hipi plus *Taq* polymerase. The PCR amplification protocol was as follows: preheating at 95 °C for 3 min, denaturation at 95 °C for 30 s, followed by 10 cycles of annealing at 72 °C (decreasing by 1 °C per cycle) for 30 s, extension at 72 °C for 45 s, then 20–30 cycles of 95 °C for 30 s, 62 °C for 30 s, and 72 °C for 45 s, concluding with a final extension of 5 min at 72 °C. The Index PCR was specifically designed to amplify the dual-index structure. During this amplification process, Illumina i5 and i7 adapters were incorporated at the 3′ and 5′ ends, respectively, using a dual indexing and adapter kit (Illumina, San Diego, CA, USA). Subsequently, the index PCR product was purified using the LaboPass PCR clean-up kit (Cosmo Genetech, Seoul, Republic of Korea). The PB buffer was added to the product and thoroughly mixed. This solution was then transferred to the provided column. A centrifugation step was carried out at 13,000 rpm for 1 min. The column was then washed with NW buffer. After discarding the eluent, the PCR product was further purified using 20 μL of EB buffer. The purified amplicons were directly sequenced according to a previously described method [57].

### 3.6. Artificial Fertilization of Edited Plants

Following the confirmation of gene editing through PCR and deep sequencing, the plants were transplanted into 10 cm pots filled with soil and relocated to an LMO glass house at the National Institute of Horticultural and Herbal Science (NIHHS), Wanju, Korea, from 2020 to 2022. The edited plants (E_0_) were nurtured under normal daylight conditions with a set temperature of 22/19 °C. Both E_0_ and E_1_ plants were self-pollinated in 2020 and 2021, respectively, to yield seeds. These E_1_ and E_2_ seeds were utilized for the in vitro UPR-related gene experiment, then grown in the LMO glass house for the next generation.

### 3.7. Tunicamycin (TM) Treatment

Surfaces of wild-type and *hy5* mutant seeds were submerged in 70% ethanol for 1 to 2 min, then agitated in a 2% NaOCl solution at 100 rpm for 10 min. Subsequently, the seeds were rinsed thoroughly four to five times with sterilized water to remove any remaining NaOCl. The cleaned seeds were stratified at 4 °C for 3 d and plated on 1 × MS medium, pH 5.8 (Duchefa Biochemie BV, Haarlem, Netherlands), supplemented with 3% sucrose and 0.25% gellan gum (PhytoTechnology Laboratories., Lenexa, KS). The plants were cultivated on MS medium with or without TM (Sigma-Aldrich Co., St. Louis, MO, USA) at the specified concentrations in a growth chamber maintained at 22 °C under long day conditions (16/8 h light/dark photoperiod, 100–200 μmol/m^2^/s photon flux density, and 60–70% relative humidity). The plant phenotypes, ROS staining, and the expression of UPR- and cell-death-related genes were compared across the plants. To measure the fresh weight, whole plants were carefully separated from the agar medium. To remove any residual agar from the roots, they were gently shaken or briefly immersed in water, allowing the medium to dislodge. After separation, the plant was gently lifted, and any excess moisture was blotted with a fresh paper towel or a similar absorbent material. Fresh weight of the whole plants was then accurately recorded.

### 3.8. Nitroblue Tetrazolium (NBT) and 3,3′-Diaminobenzidine (DAB) Staining

The nitroblue tetrazolium (NBT, Sigma-Aldrich Co., St. Louis, USA) and 3,3′-diaminobenzidine (DAB, Sigma-Aldrich Co., St. Louis, USA) staining methods were adapted for in situ detection of O_2_^−^ and H_2_O_2_, respectively [51,58,59]. Chinese cabbage seedlings were grown on normal growth media and on media treated with 150 ng/mL TM for 7 d. Untreated seedlings, grown under identical conditions, served as experimental controls.

For NBT staining, plants were immersed and vacuum-infiltrated with 0.1% NBT staining solution in a potassium phosphate buffer (50 mM) containing 10 mM sodium azide (NaN_3_). The tubes were wrapped with foil and incubated for 6 h at room temperature. Post-incubation, the stained plants were bleached in an acetic acid–glycerol–ethanol (1/1/3, *v*/*v*/*v*) solution at 100 °C for 5 min. The plants were then stored in a glycerol–ethanol (1/4, *v*/*v*) solution until photographic documentation was completed. O_2_^−^ was visualized as a blue color produced by NBT precipitation.

For DAB staining, DAB was dissolved in H_2_O, and the pH was adjusted to 3.8 using KOH. The DAB solution was freshly prepared to prevent auto-oxidation. Similar to the NBT procedure, Chinese cabbage seedlings were grown on media treated with 150 ng/mL TM for 7 d, with untreated seedlings serving as experimental controls. The plants were immersed and vacuum-infiltrated with a 1 mg/mL DAB staining solution. Then, the tubes were foil-wrapped and incubated for 6 h at room temperature. After incubation, the stained plants were bleached in an acetic acid–glycerol–ethanol (1/1/3, *v*/*v*/*v*) solution at 100 °C for 5 min, and then stored in a glycerol–ethanol (1/4, *v*/*v*) solution until photographs were taken. H_2_O_2_ was visualized as a brown color due to DAB polymerization. Images obtained after NBT and DAB staining were quantified using ImageJ software.

### 3.9. Quantitative Real-Time Polymerase Chain Reaction (qRT-PCR) Analysis

Total RNA was isolated from whole Chinese cabbage seedlings using a NucleoSpin RNA Plant kit (Macherey-Nagel, Düren, Germany) according to the manufacturer’s protocol. For each sample, 1 μg of purified RNA was used for first-strand cDNA synthesis using a ReverTra Ace-α kit (Toyobo, Osaka, Japan), following the manufacturer’s instructions. The synthesized cDNA samples were diluted (1:50) with sterile diethylpyrocarbonate-treated water. qRT-PCR was performed with a CFX96 real-time PCR system (Bio-Rad Laboratories, Hercules, CA, USA). The analysis was conducted in a 10 μL reaction volume, including 0.5 μL of each primer (10 pmol), 4 μL of template cDNA, and 5 μL of iQ™ SYBR Green Supermix (Bio-Rad Laboratories, Hercules, CA, USA). The thermal profile used was as follows: 1 cycle of 50 °C for 2 min and 95 °C for 5 min, followed by 40 cycles of 95 °C for 10 s and 60 °C for 30 s. Finally, a melting curve analysis (1 cycle) from 65 °C to 95 °C was carried out. Data from triplicates were analyzed using CFX Maestro software (Bio-Rad Laboratories, Hercules, CA, USA). The *TUBULIN* gene was amplified and used as an internal positive control. mRNA expression levels were calculated using the 2^−ΔΔCt^ method after normalization to the expression of the *TUBULIN* gene. Using sequences of UPR- and cell-death-related genes from Arabidopsis as a reference, we conducted a BRAD BLAST search to obtain sequence information for the corresponding genes in Chinese cabbage. Based on these sequences, we designed primers for qRT-PCR (Table 2 and Table 3). The primers used in this study are listed in Table 3.

### 3.10. Statistical Analysis

All values are presented as means ± SEM. Statistical analyses were conducted using GraphPad Prism (version 5.03). Data sets, including fresh weight and those with more than two groups, were analyzed using a one-way ANOVA followed by Tukey’s multiple comparison test. A significance level of *p* ≤ 0.05 was deemed statistically significant.

## 4. Conclusions

Addressing the escalating challenges of climate change requires the development of climate-resilient crops. New Breeding Techniques (NBTs), especially genome editing, have emerged as promising tools to breed crop varieties with heightened tolerance to environmental stresses. Among the species subjected to such advancements is *Brassica rapa*, where high-efficiency Agrobacterium-mediated transformations have been widely conducted [60,61,62]. Indeed, numerous studies confirm the feasibility of engineering Chinese cabbage using these methods [60,61,62,63,64]. In our study, we employed genome editing on the Chinese cabbage cultivar ‘Seoul’, known for its high transformation efficiency. From approximately 3000 cultured explants, we identified 19 plants with confirmed gene editing. The subsequent generation showcased plants with a significant increase in gene-editing frequency. Notably, we achieved plants characterized by a high editing rate and homozygosity within the target region. In the study presented, the potential role of the *HY5* gene in conferring resilience to ER stress in Chinese cabbage was investigated. The *HY5* gene, previously associated with photomorphogenesis, has increasingly been implicated in a variety of environmental stress responses, including salinity, drought, cold, and, notably, ER stress [23,38,47,48,49]. This multifunctionality of the *HY5* gene suggests its pivotal role in the overall stress physiology of plants. Its involvement in hormone signaling pathways further strengthens its candidacy as a key regulatory gene in response to changing environmental conditions.

Our study provides compelling evidence for the role of HY5 in mitigating ER stress responses in Chinese cabbage, broadening its known functions in photomorphogenesis and hormone signaling. Utilizing CRISPR/Cas9 technology, we successfully introduced loss-of-function mutations in the *HY5* gene, subsequently examining the consequences of these mutations on ER stress-induced growth inhibition in Chinese cabbage plants. The findings indicated that HY5 deficiency could indeed confer resistance to ER stress, as demonstrated by the reduced growth inhibition in *hy5* mutant lines when subjected to TM-induced ER stress (Figure 3C,D). This builds upon previous work in Arabidopsis, which reported similar resistance to ER stress in *hy5* mutants [38]. The mitigation of growth inhibition observed in our *hy5* mutant lines suggests the potential of *HY5* as a target gene to enhance stress tolerance in crops and vegetables, particularly in the light of global warming and shifting climate conditions.

The HY5-deficient lines interestingly exhibited decreased generation of ROS under ER stress (Figure 4A,B). ROS are integral to plant stress responses, and their reduced levels may partially explain the enhanced ER stress tolerance observed in the *hy5* mutants. This finding is particularly intriguing as it hints at the possibility of involvement of HY5 in regulating antioxidant defense systems, an avenue that warrants further exploration. In addition to alterations in ROS levels, we found that the *HY5* mutations were associated with decreased expression of UPR- and cell-death-related genes under ER stress (Figure 5). These results suggest that HY5 could be a key player in ER stress responses, potentially regulating the expression of these genes and thus influencing cell survival under stress conditions. However, in the absence of TM, the expression of these genes in *hy5* mutants remained similar to wild-type lines, indicating that HY5 may predominantly function in stress responses rather than in regular cellular processes (Figure 3C,D). Nevertheless, the precise mechanisms surrounding the involvement of HY5 in ER stress response and its regulation of the UPR and cell death-related genes remain to be fully elucidated. To further expand our knowledge of the underlying molecular processes, it will be crucial for future research to dissect the complex interplay between HY5, ROS generation, and the regulation of UPR- and cell-death-related genes under circumstances of ER stress.

The findings from this study not only advance our understanding of plant stress physiology but also have practical implications. Developing *HY5* mutant lines of Chinese cabbage, and potentially other crops, can offer valuable tools for farmers dealing with the adversities of climate change. Furthermore, since ER stress has been implicated in affecting crop yield and quality, harnessing the potential of *HY5* mutations can contribute significantly to food security in the face of global warming. While the current study sheds light on the potential of *HY5* mutations in conferring stress tolerance, further studies are needed to unravel the broader physiological, biochemical, and molecular changes in these mutants under various stress conditions. Additionally, exploring the potential role of *HY5* in other crops can further validate its universality as a master regulator of stress responses. In conclusion, this study underscores the pivotal role of the *HY5* gene in plant stress responses and the potential of genome editing techniques in developing climate-resilient crops. As the challenges posed by climate change intensify, such innovations are bound to play a crucial role in shaping the future of agriculture.

## Figures and Tables

**Figure 1 ijms-24-13105-f001:**
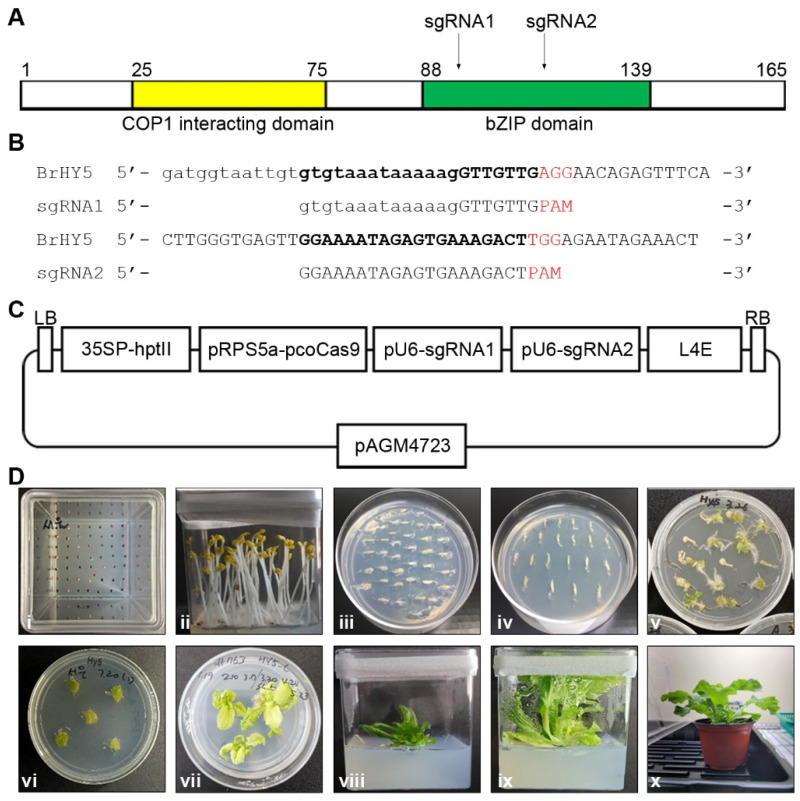
Representation of HY5 protein structure, sgRNA design, and generation of transgenic Chinese cabbage bearing the sgRNA construct. (**A**) Diagram depicting the structure of the HY5 protein in Chinese cabbage and the target sites. The yellow box illustrates the COP1 interacting domain, while the green box signifies the bZIP domain. The black arrow points to the sgRNA target sites. (**B**) Schematic representation of the sgRNA sequences. The exon sequence is displayed in uppercase letters, and the intron sequence is in lowercase letters. The target sequence is emphasized in bold, and the PAM sequence is highlighted in red. (**C**) Diagram of the pAGM4723-HY5 bZIP sgRNA vector. 35SP: 35S promoter; hptII: hygromycin B phosphotransferase; pRPS5a: ribosomal protein subunit 5a promoter; pcoCas9: plant codon-optimized Cas9; pU6: Arabidopsis U6 promoter; L4E: linker. (**D**) Process of Agrobacterium-mediated Chinese cabbage transformation and generation of transgenic Chinese cabbage bearing the sgRNA construct: (**i**) in vitro seed germination; (**ii**) in vitro Chinese cabbage seedlings; (**iii**) pre-cultivation of hypocotyls in media in vitro; (**iv**) hypocotyls inoculated with Agrobacterium and incubated in co-culture media in vitro; (**v**) induction of callus at both ends of the hypocotyls; (**vi**) callus formation from callus-inducing selection media; (**vii**) regeneration of shoots from calli under callus-inducing selection media; (**viii**) root induction from the regenerated shoot in root-inducing media; (**ix**) in vitro regeneration of plants; (**x**) growth of T_0_ transgenic plants post-transplantation into soil-filled pots in vivo.

**Figure 2 ijms-24-13105-f002:**
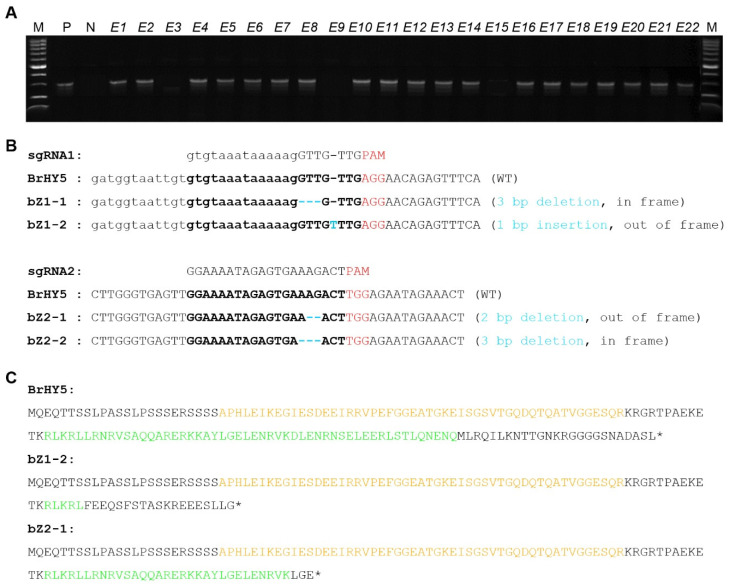
Knockout of *HY5* achieved through CRISPR/Cas9-induced mutagenesis. (**A**) Detection of sgRNA via genomic DNA PCR. M: 100 bp size marker, P: positive control, N: negative control, *E1*~*E22*: the mutant lines. (**B**) Illustration of representative *hy5* mutant genotypes. Target sequences are highlighted in bold, with PAM sequences in red. Deletions are represented by a blue dash, and insertions are indicated by blue letters. (**C**) Diagram of HY5 amino acid sequences. The COP1 interacting domain is denoted by yellow letters, and the bZIP domain is indicated in green. * indicates amino acid termination.

**Figure 3 ijms-24-13105-f003:**
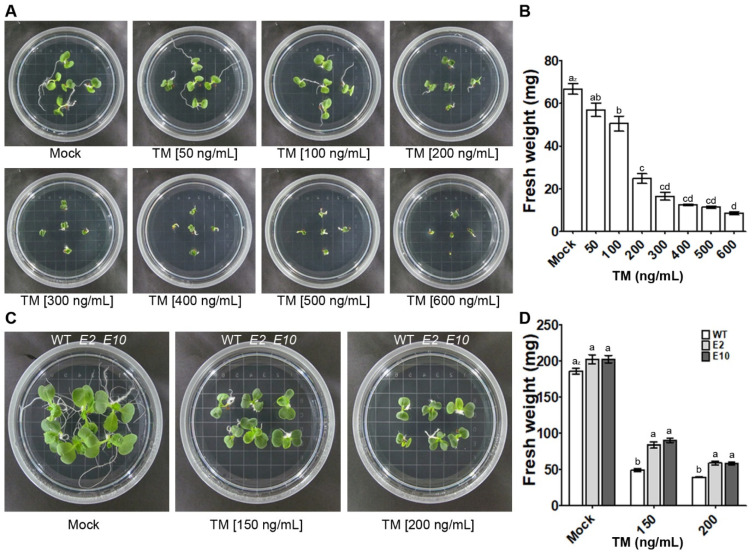
The *hy5* mutants display diminished growth inhibition under ER stress conditions. (**A**) Images of 5 d old wild-type plants grown either on MS medium alone (Mock) or on MS medium supplemented with TM (50–600 ng/mL). (**B**) Fresh weight of 5 d old wild-type plants grown either on MS medium alone (Mock) or on MS medium supplemented with TM (50–600 ng/mL). Data represent the means ± SE of five independent experiments. Different letters (*p* < 0.05, one-way ANOVA) denote significant differences among different treatments. (**C**) Images of 7 d old wild-type and *hy5* mutant plants grown either on MS medium alone (Mock) or on MS medium supplemented with TM (150 and 200 ng/mL). (**D**) Fresh weight of 7 d old wild-type and *hy5* mutant plants grown either on MS medium alone (Mock) or on MS medium supplemented with TM (150 and 200 ng/mL). Data represent the means ± SE of four independent experiments. Different letters (*p* < 0.05, one-way ANOVA) denote significant differences among different treatments.

**Figure 4 ijms-24-13105-f004:**
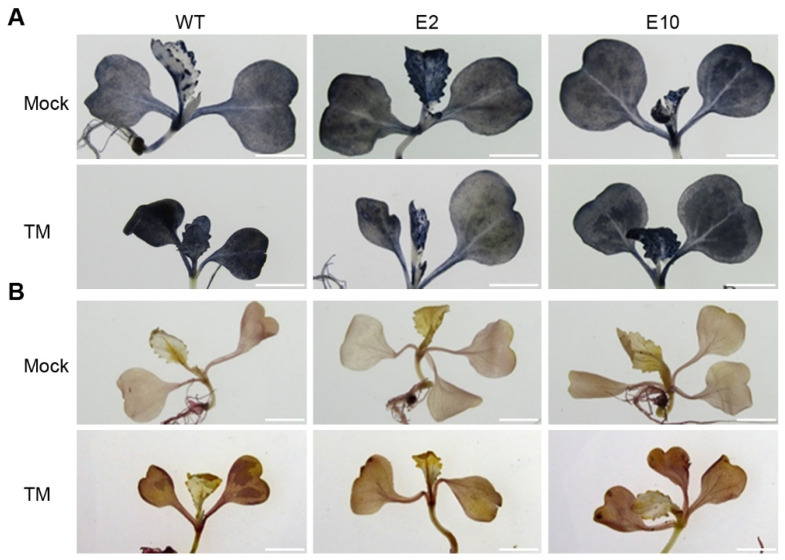
The *hy5* mutants display reduced ROS levels under ER stress conditions. (**A**,**B**) Visualization of superoxide radicals (O_2_^−^) and hydrogen peroxide (H_2_O_2_) in the leaf tissues of 7 d old wild-type and *hy5* mutant plants. Plants were grown either on MS medium alone (Mock) or on MS medium supplemented with TM (150 ng/mL). ROS were detected using (**A**) nitroblue tetrazolium (NBT) for superoxide radicals and (**B**) 3,3′-diaminobenzidine (DAB) for hydrogen peroxide, respectively. Scale bars represent 1 cm.

**Figure 5 ijms-24-13105-f005:**
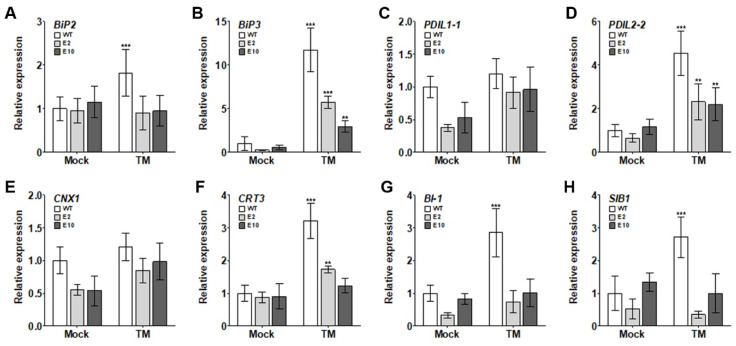
Lower expression of UPR- and cell death-related genes in *hy5* mutants under ER stress. (**A**–**H**) Expression profiles of UPR-related genes (*BiP2*, *BiP3*, *PDIL1-1*, *PDIL2-2*, *CNX1*, and *CRT3*) along with cell-death-related genes (*BI-1* and *SIB1*) as analyzed via qRT-PCR. Plants were grown as per the conditions mentioned in Figure 3C (Mock and TM 150 ng/mL). The *TUBULIN* gene served as the reference gene for normalizing expression levels. Data illustrate the means ± SE derived from three separate experiments. The asterisks denote significant differences (** *p* ≤ 0.01, *** *p* ≤ 0.001) ascertained by one-way analysis of variance (ANOVA).

**Table 1 ijms-24-13105-t001:** Mutation rates detected in the *HY5* gene by targeted deep sequencing.

		Nucleotide Sequence (5′->3′)	Type	Mutant Frequency (%)	Reads Number	Total Reads Number
*E2*	gRNA1	GTGTAAATAAAAAGGTTGTTGAGG	WT	4.38	1337	30,547
		GTGTAAATAAAAAG– –GTTGAGG	−3	62.3	19,039	
		GTGTAAATAAAAAGGTTG**T**TTGAGG	+1	33.3	10,171	
	gRNA2	GGAAAATAGAGTGAAAGACTTGG	WT	7.91	3775	47,718
		GGAAAATAGAGTGAA– –ACTTGG	−2	57.5	27,429	
		GGAAAATAGAGTGA– –ACTTGG	−3	34.6	16,514	
*E10*	gRNA1	GTGTAAATAAAAAGGTTGTTGAGG	WT	2.7	946	34,503
		GTGTAAATAAAAAG– –GTTGAGG	−3	62.2	21,454	
		GTGTAAATAAAAAGGTTG**T**TTGAGG	+1	35.1	12,103	
	gRNA2	GGAAAATAGAGTGAAAGACTTGG	WT	2.9	1320	46,112
		GGAAAATAGAGTGAA– –ACTTGG	−2	56.3	25,961	
		GGAAAATAGAGTGA– –ACTTGG	−3	40.8	18,831	

The PAM sequence is indicated by an underline. Deletion is represented by a dash. Insertion is denoted by bold letters.

**Table 2 ijms-24-13105-t002:** List of UPR- and cell-death-related genes of *Arabidopsis thaliana* and corresponding genes of *Brassica rapa*.

Gene	Arabidopsis	Gene	*Brassica rapa*	Gene Annotation
*CNX1*	AT5G61790	*BrCNX1*	Bra035913	Calnexin 1, unfolded protein binding
*CRT3*	AT1G08450	*BrCRT3*	Bra031627	Calreticulin 3, unfolded protein binding
*BiP2*	AT5G42020	*BrBiP2*	Bra015047	Heat shock protein 70 (Hsp 70) family protein
*BiP3*	AT1G09080	*BrBiP3*	Bra031657	Heat shock protein 70 (Hsp 70) family protein
*PDIL1-1*	AT1G21750	*BrPDI1-1*	Bra016405	PDI-like 1-1, protein disulfide isomerase activity
*PDIL2-2*	AT1G04980	*BrPDI2-2*	Bra015375	PDI-like 2-2, protein disulfide isomerase activity
*BI-1*	AT5G47120	*BrBI-1*	Bra022106	BAX inhibitor 1, Functions as an attenuator of biotic and abiotic types of cell death
*SIB1*	AT3G56710	*BrSIB1*	Bra007265	sigma factor binding protein 1, defense responses
*TUB*	AT5G12250	*BrTUB*	Bra008903	beta-6 tubulin, internal control

**Table 3 ijms-24-13105-t003:** List of primers and their sequences.

Primer Name	Sequence (5′ to 3′)	Gene
*CNX1* F	ATCCCTGACAAGACCATCC	*BrCNX1*
*CNX1* R	CCTCCCACATACCATCTTCC	*BrCNX1*
*CRT3* F	TCTTCTCTCTTCTCACTCTCAC	*BrCRT3*
*CRT3* R	CTTTATTGTCAGGATCGCCG	*BrCRT3*
*BiP2* F	GGGAAGCCGTACATTCAAG	*BrBiP2*
*BiP2* R	GAGACCAGCAATAACACCAG	*BrBiP2*
*BiP3* F	CTGACTTCTCTGAGCCTTTAAC	*BrBiP3*
*BiP3* R	TCTTCACCACCTTCACCAC	*BrBiP3*
*PDIL1-1* F	AAAATCTCAACCCATCCCAACC	*BrPDIL1-1*
*PDIL1-1* R	ATGACACAGCGACTTCGTCC	*BrPDIL1-1*
*PDIL2-2* F	CTTGAAGCCAATGCTGGAC	*BrPDIL2-2*
*PDIL2-2* R	ACCATATCCTCCAACTCCAAC	*BrPDIL2-2*
*BI-1* F	CATCCTCATCACTGCGTTTG	*BrBI-1*
*BI-1* R	AGTGTCCACCACCATGTATC	*BrBI-1*
*SIB1* F	TTCAGCCAACAAAGCCATC	*BrSIB1*
*SIB1* R	AACATCTCTTCACCATCCAAC	*BrSIB1*
*TUB* F	TGACTGTCTTCAGGGTTTCCA	*TUBULIN*
*TUB* R	CACCGTGTCCGAGACCTTAG	*TUBULIN*

## Data Availability

Not applicable.

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
