# Peer review of "CRISPR/Cas9-Mediated HY5 Gene Editing Reduces Growth Inhibition in Chinese Cabbage (Brassica rapa) under ER Stress"

_ijms, 2023, doi:10.3390/ijms241713105_

Round 1
Reviewer 1 Report
Title must be “CRISPR/Cas9-mediated HY5 gene editing reduces growth inhibition in Chinese cabbage (Brassica rapa) under ER stress”
This study successfully demonstrated potential of genome editing for improving stress tolerance in Brassica rapa using CRISPR/Cas9-mediated targeted mutation of HY5 gene. This is a small good piece of work. This study developed hy5 mutant plants and the mutants displayed less severe growth inhibition compared to the wild type due to lower level of ROS in mutants using NBT and DAB staining methods. Importantly, this study studied the relative expression of UPR- and cell death-related genes. The hy5 mutant plants exhibited lower expression levels of UPR- and cell death-related genes than the wild type. Author can avoid the generalising and extrapolating the statement on mutation of HY5 for developing climate-resilient crops and vegetables.
Overall, manuscript content is interest to the readers. Typographical errors throughout the text requires to be corrected. Include country name wherever product brand name is given. However, I have some suggestions to improve the presentation of this manuscript.
Abstract: This section needs to be concised the content especially lines 27-29.
Line 29, HYPOCOTYL 5 (HY5) gene and its influence ….
Line 29, sgRNA was designed to…
Line 31, Chinese cabbage to endoplasmic reticulum (ER) stress, plants were treated with tunicamycin
Line 34, Using NBT (Expand it) and DAB (Expand it) for the first instance in the abstract.
Introduction: This section gives a clear background on functional role of Hy5 genes based on available information.
Materials and methods:
Line 124, 2% sodium hypochlorite (NaOCl)
Line 127, in vitro must be in italiucs
Line 128, include the relative humidity and light intensity maintained.
Line 172-173, sgRNA primer sequence is not matching with the sequence given in Table 1. There were two sgRNA used but only one pair is given that too not matching. Look into it.
Line 182, were used for the first round of PCR designed…
Line 184-187, There were two annealing temperature for one pair of primers. This section is not clear. Appropriate changes are required.
Line 199, seeds were utilized for the in vitro
Line 201, Expand the TM, Importance of TM required to be included in the introduction section for easy understanding of readers.
Lines 122-129 and 203-210, there are two surface seed sterilization protocols followed in this manuscript. This should be a common protocol.
Line 212, Nitroblue Tetrazolium (NBT) and 3,3’-diaminobenzidine (DAB) Staining
Line 214, were adapted for in situ detection
Line 225, were grown on media treated with 150 ng/mL TM for 7 d and untreated
Result section:
Line 298-312, This study generated only 19 edited lines, and mutation frequency extrapolation is too high. With minimal number of events and extrapolation of different combinations of mutants might mislead the data. This section required to be revised and shortened.
Line 315-316, various environmental stress factors such as heat, drought, salinity, herbicide application, and pathogen infection are known to affect the protein folding capability in the ER of plant cells, avoid the repletion of this sentence. It is repeated in many occasions in the manuscript.
Line 349, can reduce ROS production and thus stress tolerance in Chinese cabbage under ER stress
Line 357-363, This is mixed with materials and methods part here.
This study could have studied the unfolded protein concentration in ER. This could have been much interested to the readers.
Discussion:
Short discussion. Needs to be expanded based on results.
Conclusion part is general statement. This should be concise part of the main results obtained in this study. And also futuristic view cam be given.
Requires improvement.
Author Response
Abstract:
This section needs to be concised the content especially lines 27-29.
- Thank you for the reviewer's comment. Based on the reviewer's suggestion, the content has been condensed. Please refer to the revised abstract.
Line 29, HYPOCOTYL 5 (HY5) gene and its influence ….
- Thank the reviewer for pointing this out. According to the reviewer's suggestion, the content has been revised. Please refer to the revised abstract.
Line 29, sgRNA was designed to…
- Thank the reviewer for pointing this out. According to the reviewer's suggestion, the content has been revised. Please refer to the revised abstract.
Line 31, Chinese cabbage to endoplasmic reticulum (ER) stress, plants were treated with tunicamycin
- Thank the reviewer for pointing this out. According to the reviewer's suggestion, the content has been revised. Please refer to the revised abstract.
Line 34, Using NBT (Expand it) and DAB (Expand it) for the first instance in the abstract.
- Thank the reviewer for pointing this out. According to the reviewer's suggestion, the content has been revised. Please refer to the revised abstract.
Introduction:
This section gives a clear background on functional role of HY5 genes based on available information.
Materials and methods:
Line 124, 2% sodium hypochlorite (NaOCl)
- Thank the reviewer for pointing this out. According to the reviewer's suggestion, the content has been revised. Please refer to the revised sentence in the Materials and methods section.
Line 127, in vitro must be in italics
- Thank the reviewer for pointing this out. According to the reviewer's suggestion, the content has been revised. Please refer to the revised sentence in the Materials and methods section.
Line 128, include the relative humidity and light intensity maintained.
- Thank the reviewer for pointing this out. According to the reviewer's suggestion, the content has been revised. Please refer to the revised sentence in the Materials and methods section.
Line 172-173, sgRNA primer sequence is not matching with the sequence given in Table 1. There were two sgRNA used but only one pair is given that too not matching. Look into it.
- Thank the reviewer for pointing this out. According to the reviewer's suggestion, the content has been revised. Please refer to the revised sentence in the Materials and methods section.
Line 182, were used for the first round of PCR designed…
- Thank the reviewer for pointing this out. According to the reviewer's suggestion, the content has been revised. Please refer to the revised sentence in the Materials and methods section.
Line 184-187, There were two annealing temperature for one pair of primers. This section is not clear. Appropriate changes are required.
- Thank the reviewer for pointing this out. According to the reviewer's suggestion, the content has been revised. Please refer to the revised sentence in the Materials and methods section.
Line 199, seeds were utilized for the in vitro
- Thank the reviewer for pointing this out. According to the reviewer's suggestion, the content has been revised. Please refer to the revised sentence in the Materials and methods section.
Line 201, Expand the TM, Importance of TM required to be included in the introduction section for easy understanding of readers.
- We thank the reviewer for pointing this out. Based on the reviewer's suggestion, the content has been updated and the importance of TM is now highlighted in the introduction section. Please refer to the revised sentences in both the 'Materials and Methods' and 'Introduction' sections.
Lines 122-129 and 203-210, there are two surface seed sterilization protocols followed in this manuscript. This should be a common protocol.
- Thank the reviewer for pointing this out. According to the reviewer's suggestion, the content has been revised. Please refer to the revised sentence in the Materials and methods section.
Line 212, Nitroblue Tetrazolium (NBT) and 3,3’-diaminobenzidine (DAB) Staining
- Thank the reviewer for pointing this out. According to the reviewer's suggestion, the content has been revised. Please refer to the revised sentence in the Materials and methods section.
Line 214, were adapted for in situ detection
- Thank the reviewer for pointing this out. According to the reviewer's suggestion, the content has been revised. Please refer to the revised sentence in the Materials and methods section.
Line 225, were grown on media treated with 150 ng/mL TM for 7 d and untreated
- Thank the reviewer for pointing this out. According to the reviewer's suggestion, the content has been revised. Please refer to the revised sentence in the Materials and methods section.
Result section:
Line 298-312, This study generated only 19 edited lines, and mutation frequency extrapolation is too high. With minimal number of events and extrapolation of different combinations of mutants might mislead the data. This section required to be revised and shortened.
- We thank the reviewer for their insightful suggestion. Based on this feedback, we have incorporated the relevant information in the Conclusions section. Please refer to the updated sentence therein.
Line 315-316, various environmental stress factors such as heat, drought, salinity, herbicide application, and pathogen infection are known to affect the protein folding capability in the ER of plant cells, avoid the repletion of this sentence. It is repeated in many occasions in the manuscript.
- Thank you for pointing this out. Based on the reviewer's suggestion, the content has been revised to avoid repetition. Please refer to the updated results.
Line 349, can reduce ROS production and thus stress tolerance in Chinese cabbage under ER stress
- Thank you for pointing this out. Based on the reviewer's suggestion, the content has been revised. Please refer to the updated results.
Line 357-363, This is mixed with materials and methods part here.
- Thank you for the reviewer's comments. Based on the reviewer's suggestion, some sentences have been moved to the 'Materials and Methods' section, and the 'Results' section has been modified. Please refer to the updated 'Materials and Methods' and 'Results' sections.
This study could have studied the unfolded protein concentration in ER. This could have been much interested to the readers.
- In this study, we did not measure the unfolded protein concentration in the ER. However, by measuring the expression of UPR- and cell death-related genes, we were able to confirm the occurrence of UPR due to ER stress. We kindly ask for the reviewer's understanding regarding this aspect and hope to have the opportunity to directly measure the unfolded protein concentration in the ER in future research.
Discussion:
Short discussion. Needs to be expanded based on results. Conclusion part is general statement. This should be concise part of the main results obtained in this study. And also futuristic view cam be given.
- Thank you for the reviewer's kind comments. Following the reviewer's suggestions, we have expanded the 'Discussion' section based on the results. We have also drawn a concise conclusion based on the main results obtained in this study and provided a perspective on the future. Please refer to the updated 'Discussion' section.
Reviewer 2 Report
Ye-Rin Lee and colleagues present a study on CRISPR/Cas9-mediated editing of the HY5 gene and its effect on growth inhibition in Chinese cabbage. The topic is of interest to the readership since it investigates the role of HY5 in plants, opening doors for the exploitation of this gene to enhance stress tolerance across a broad range of crop and vegetable species. However, before the manuscript can be published, a revision of the structure and content is necessary.
Remarks:
1. Introduction. The introduction is too long, this is not a textbook, shorten the information in the lines 82-85, please. The authors should explain why the examples of transgenic plants are given in the lines 104-110? It is necessary to give some examples of genome edited plants.
Why did you choose the ER stress and reactive oxygen species as stress indicator? Why do you use a tunicamycin as a source of stress? At the same time, one of the target genes for determining the expression level is the heat-shock protein. It is necessary to give an explanation.
2. Materials and methods.
- For each instrument should mention the model, manufacturer, and country. It is also necessary to describe for each name of reagent kit.
- Primer sequences and conditions of second round of PCR are not described (line 187).
- PCR product purification protocol and amplification protocol with barcodes are not described (line 188).
- What part of the sgRNA was targeted by the primers? How did you weed out false-positive results and control the absence of stem-loop (line 172)?
- How was raw plant biomass studied? How was the agar medium disposed of on the plant roots (should be described in the materials and methods)? Did the development of the above-ground biomass or root system change when the tunicamycin was introduced?
- It is necessary to mention which program you used when measuring the intensity of plant coloring (for in situ detection of O2− and H2O2).
- It is necessary to describe how you calculated the genes relative expression. Have you used the 2-∆∆Ct method? One-way ANOVA is not enough to assess significant difference between variants (in all analyses). Please mention which method you used for comparing (t-test? LSD-test?). Mention which program you used for data analysis and visualization. An additional subsection is needed in materials and methods.
2. Results. Why text rewired climate change and known results (lines 248-257) can be found in the Result part? The authors should focus this part on the results of their study.
Moreover, the authors included in the section Results the discussion of the obtained data- (lines 315-317, 325, 339, 353 and so on). Authors should carefully compare their results with the literature data and separate them into Discussion section.
3. Discussion. On the contrary, the authors should rename the Discussion section to the Сconclusion, since in terms of its content it is a conclusion.
4. Technical notes:
Line 323 - (A, B). indicate the Fig. number.
Author Response
- Introduction.
The introduction is too long, this is not a textbook, shorten the information in the lines 82-85, please. The authors should explain why the examples of transgenic plants are given in the lines 104-110? It is necessary to give some examples of genome edited plants.
- Thank you for the reviewer's meticulous feedback. In response to the suggestions, we have condensed and revised the 'introduction' section. Instead of providing examples of transgenic plants, we have included examples of genome-edited plants. Please refer to the updated 'introduction'.
Why did you choose the ER stress and reactive oxygen species as stress indicator? Why do you use a tunicamycin as a source of stress? At the same time, one of the target genes for determining the expression level is the heat-shock protein. It is necessary to give an explanation.
- Thank you for your detailed feedback. We chose the UPR caused by ER stress and reactive oxygen species as stress indicators because they manifest or arise in response to multiple stresses. Tunicamycin (TM) is a natural antibiotic that induces ER stress in cells by inhibiting the early phase of N-linked glycan synthesis in proteins. This interruption leads to the accumulation of misfolded proteins, triggering the UPR in plants (Koizumi et al., 1999- 10517826). While the target genes are chaperones located in the endoplasmic reticulum, they are also representative UPR marker genes. We have added and revised this information at the end of the second paragraph and the beginning of the third paragraph. Please refer to the updated 'introduction'.
- Materials and methods.
For each instrument should mention the model, manufacturer, and country. It is also necessary to describe for each name of reagent kit.
- Thank the reviewer for pointing this out. According to the reviewer's suggestion, the contents have been revised. Please refer to the revised sentences in the Materials and methods section.
Primer sequences and conditions of second round of PCR are not described (line 187).
- We thank the reviewer for pointing this out. The first and second rounds of PCR were performed under the same conditions. In response to the reviewer's suggestion, we have made revisions. Please refer to the updated sentences in the 'Materials and Methods' section.
PCR product purification protocol and amplification protocol with barcodes are not described (line 188).
- We thank the reviewer for pointing this out. We have revised and added the necessary information to the "Materials and Methods" section. Please refer to the updated content in the "Materials and Methods" section.
What part of the sgRNA was targeted by the primers? How did you weed out false-positive results and control the absence of stem-loop (line 172)?
- Thank you to the reviewer for the important indications. Based on the reviewer's feedback, we have added the necessary information to the "Materials and Methods" section. Please refer to the updated content in the "Materials and Methods" section.
How was raw plant biomass studied? How was the agar medium disposed of on the plant roots (should be described in the materials and methods)? Did the development of the above-ground biomass or root system change when the tunicamycin was introduced?
- Thank the reviewer for pointing this out. Under our experimental conditions, treating plants with tunicamycin affected both the development of the above-ground biomass and the root system. Therefore, we measured the fresh weight of the whole plants. According to the reviewer's suggestion, the content has been revised. Please refer to the revised sentences in the Materials and methods section.
It is necessary to mention which program you used when measuring the intensity of plant coloring (for in situ detection of O2− and H2O2).
- Thank the reviewer for pointing this out. According to the reviewer's suggestion, the content has been revised. Please refer to the revised sentence in the Materials and methods section.
It is necessary to describe how you calculated the genes relative expression. Have you used the 2-∆∆Ct method? One-way ANOVA is not enough to assess significant difference between variants (in all analyses). Please mention which method you used for comparing (t-test? LSD-test?). Mention which program you used for data analysis and visualization. An additional subsection is needed in materials and methods.
- Thank the reviewer for pointing this out. According to the reviewer's suggestion, the content has been revised. Please refer to the revised sentences in the Materials and methods section.
- Results.
Why text rewired climate change and known results (lines 248-257) can be found in the Result part? The authors should focus this part on the results of their study. Moreover, the authors included in the section Results the discussion of the obtained data- (lines 315-317, 325, 339, 353 and so on). Authors should carefully compare their results with the literature data and separate them into Discussion section.
- Thank you to the reviewer for pointing this out. Based on the reviewer's suggestion, the content has been revised. However, we kindly ask for your understanding as discussion has been retained to ensure clarity and ease of comprehension for readers. Thus, we have renamed the 'Results' section to 'Results and Discussion' section. Please refer to the updated sentences in the 'Results and Discussion' section.
- Discussion.
- On the contrary, the authors should rename the Discussion section to the conclusion, since in terms of its content it is a conclusion.
- Thank you to the reviewer for pointing this out. Based on the reviewer's suggestion, we have renamed the 'Discussion' section to 'Conclusions' section. Please refer to the updated sentences in the 'Conclusions' section.
- Technical notes:
Line 323 - (A, B). indicate the Fig. number.
- Thank the reviewer for pointing this out. According to the reviewer's suggestion, the content has been revised. Please refer to the revised results.
Round 2
Reviewer 2 Report
All comments taken into account in the new version of the manuscript.